

# Isoscapes of remnant and restored Hawaiian montane forests reveal differences in biological nitrogen fixation and carbon inputs

Christopher B. Wall[1,2], Sean O. I. Swift[2], Carla M. D'Antonio[3], Gerhard Gebauer[4] and Nicole A. Hynson[2]

[1] University of California, San Diego, San Diego, CA, United States
[2] University of Hawaii at Manoa, Honolulu, HI, United States
[3] University of California, Santa Barbara, Santa Barbara, CA, United States
[4] Universität Bayreuth, Bayreuth, Germany

Corresponding author
Christopher B. Wall,
cbwall@ucsd.edu

## ABSTRACT

Deforestation and subsequent land-use conversion has altered ecosystems and led to negative effects on biodiversity. To ameliorate these effects, nitrogen-fixing ($N_2$-fixing) trees are frequently used in the reforestation of degraded landscapes, especially in the tropics; however, their influence on ecosystem properties such as nitrogen (N) availability and carbon (C) stocks are understudied. Here, we use a 30-y old reforestation site of outplanted native $N_2$-fixing trees (*Acacia koa*) dominated by exotic grass understory, and a neighboring remnant forest dominated by *A. koa* canopy trees and native understory, to assess whether restoration is leading to similar N and C biogeochemical landscapes and soil and plant properties as a target remnant forest ecosystem. We measured nutrient contents and isotope values ($\delta^{15}N$, $\delta^{13}C$) in soils, *A. koa*, and non-$N_2$-fixing understory plants (*Rubus* spp.) and generated $\delta^{15}N$ and $\delta^{13}C$ isoscapes of the two forests to test for (1) different levels of biological nitrogen fixation (BNF) and its contribution to non-$N_2$-fixing understory plants, and (2) the influence of historic land conversion and more recent afforestation on plant and soil $\delta^{13}C$. In the plantation, *A. koa* densities were higher and foliar $\delta^{15}N$ values for *A. koa* and *Rubus* spp. were lower than in the remnant forest. Foliar and soil isoscapes also showed a more homogeneous distribution of low $\delta^{15}N$ values in the plantation and greater influence of *A. koa* on neighboring plants and soil, suggesting greater BNF. Foliar $\delta^{13}C$ also indicated higher water use efficiency ($WUE_i$) in the plantation, indicative of differences in plant-water relations or soil water status between the two forest types. Plantation soil $\delta^{13}C$ was higher than the remnant forest, consistent with greater contributions of exotic $C_4$-pasture grasses to soil C pools, possibly due to facilitation of non-native grasses by the dense *A. koa* canopy. These findings are consequential for forest restoration, as they contribute to the mounting evidence that outplanting $N_2$-fixing trees produces different biogeochemical landscapes than those observed in reference ecosystems, thereby influencing plant-soil interactions which can influence restoration outcomes.

## INTRODUCTION

Forests provide essential ecosystem services including carbon (C) storage and nutrient cycling. However, human-induced disturbances, deforestation, land conversion, and invasive species have led to devastating losses in biodiversity and potentially irreversible alterations to biogeochemical processes (*Cramer et al., 2004*; *Asner et al., 2008*; *Handa et al., 2014*; *Veldkamp et al., 2020*). While active forest restoration has been touted as a tool to spur secondary succession and regain desirable ecosystem states, it remains unclear if this practice will lead to the recuperation of the biogeochemical properties provided by primary forest reference ecosystems (*Aerts & Honnay, 2011*; *Sullivan et al., 2019*; *Yelenik et al., 2021*). As land management strategies focused on restoration are developed to reconcile losses of ecosystem services, there is a need to understand the spatial extent and long-term impacts of these restoration strategies on restoring biogeochemical properties and C and nutrient cycling.

Nitrogen (N) is a key component of biogeochemical cycles, and N is often one of the most limiting nutrients for plant growth and photosynthesis—especially in early successional or secondary tropical forests (*Vitousek & Farrington, 1997*). Furthermore, the conversion of tropical forests to pasture and other agricultural crops can lead to significant changes in soil properties such rates of N cycling and N availability, as well as C sequestration and storage (*Veldkamp et al., 2020*). Trees capable of atmospheric nitrogen ($N_2$) fixation *via* symbiotic interactions with root nodule inhabiting bacteria (ex. *Rhizobia* spp.) can be integral to restoring forests by catalyzing ecological succession, reversing the effects of destructive land-use practices on soil N availability, and aiding global efforts to mitigate climate change through increasing C sequestration (*Chazdon, 2003*; *Batterman et al., 2013*; *Chazdon et al., 2016*; *Levy-Varon et al., 2019*; but see, *Kou-Giesbrecht & Menge, 2019*). Specifically, as the density of $N_2$-fixing trees, such as leguminous species in the genus *Acacia* (*Resh, Binkley & Parrotta, 2002*) increases, N in the soil and neighboring plants can also increase (*Sitters, Edwards & Olde Venterink, 2013*). Therefore, biological nitrogen fixation (BNF) has the capacity to positively influence the performance and N-budgets of both $N_2$-fixing and non-$N_2$-fixing plants.

In the tropics, $N_2$-fixing trees are often used as ecosystem engineers for the restoration of degraded landscapes (*Fisher, 1995*; *Scowcroft, Haraguchi & Hue, 2004*). However, the outcomes of these efforts are mixed, with some restored areas stalled in apparent alternative stable states (*Yelenik, 2017*), while others progress toward ecosystem targets and restoration goals (*Fisher, 1995*; *Rhoades, Eckert & Coleman, 1998*; *Koutika et al., 2021*). These differing effects of $N_2$-fixing trees may be owed to heterogeneous N inputs across landscapes (*Dixon et al., 2010*; *Sullivan et al., 2014*), as well as ecosystem-specific biotic and abiotic factors (*Pearson & Vitousek, 2002*; *Staddon, 2004*; *Wynn & Bird, 2007*; *Dixon et al., 2010*; *Barron, Purves & Hedin, 2011*; *Sitters, Edwards & Olde Venterink, 2013*).

For example, active restoration that relies on establishing forests of $N_2$-fixing trees may have unintended consequences, such as promoting weedy or invasive species (*Funk & Vitousek, 2007*), thereby altering successional trajectories toward non-target states (*Stinca*
*et al., 2015*). A better understanding of the spatiotemporal effects of $N_2$-fixing trees on ecosystems is needed, especially in the context of forest restoration.

Stable isotopes are time-integrating markers that can provide insight into biogeochemical processes and shifts in ecosystem services occurring across multiple landscapes and spatiotemporal scales (*Cheesman & Cernusak, 2016*). Spatially explicit, geo-referenced isotope landscapes (termed 'isoscapes') allow for the measured isotope values of individual replicates (*i.e.*, within or among species, or sample types) to be interpolated to landscape scales (*Bowen, 2010*). While only a few studies have used isoscapes in plant ecology (*Hellmann et al., 2011*; *Rascher et al., 2012*), their application may provide new and important insights into plant-microbe and plant-soil feedbacks, such as BNF by $N_2$-fixing trees, which can vary at relatively smaller spatial scales.

BNF results in N stable isotope values ($\delta^{15}N$) that are more similar to the atmosphere (~0‰) relative to N assimilated from soil N pools (*Craine et al., 2015*). Soil $\delta^{15}N$ values represent an integration of N inputs and outputs in an ecosystem, and are influenced by processes that lead to isotope fractionation (*i.e.*, nitrification, denitrification, ammonia volatilization), atmospheric deposition, leaching, as well as the pedogenic and environmental factors that shape these processes (*Natelhoffer & Fry, 1988*; *Austin & Vitousek, 1998*; *Martinelli et al., 1999*; *Burnett et al., 2022*). Variability in soil $\delta^{15}N$ can complicate the interpretation of leaf $\delta^{15}N$ values, especially for non-$N_2$-fixing plants (*Robinson, 2001*). However, the low $\delta^{15}N$ values associated with BNF are distinct enough to be traced through soils and vegetation (*Hellmann et al., 2011*). For instance, $\delta^{15}N$ isoscapes revealed the contribution of an invasive, non-native $N_2$-fixing shrub (*Acacia longifolia*) to the N pools of surrounding plants and the corresponding modification of nutrient cycling and community function due to this invasive species (*Rascher et al., 2012*).

Along with standard comparisons of C to N concentrations, leaf and soil C stable isotope values ($\delta^{13}C$) can provide additional evidence on the rates of nutrient cycling, the relative contribution of plant functional types (*i.e.*, $C_3$ *vs.* $C_4$ photosynthesis) to soil C pools, and the integrated plant water use efficiency ($WUE_i$) of plants in an ecosystem (*Cernusak et al., 2013*; *Driscoll et al., 2020*). Combined, $\delta^{13}C$ and $\delta^{15}N$ values sampled across a landscape can provide spatially explicit information on plant performance and ecosystem processes, which can be compared among ecological community members or between communities (*Hellmann et al., 2016*). While it is clear that isoscapes provide new perspectives on the spatial relationships of biochemical processes and biological interactions fundamental to ecology (*Cheesman & Cernusak, 2016*; *McCue et al., 2020*), they have yet to be widely applied to understand plant-soil feedbacks in the context of restoration.

Native Hawaiian montane mesic and wet forests have significantly declined due to over a century of deforestation and extensive human- and livestock-mediated disturbance (*Pau, Gillespie & Price, 2009*; *McDaniel et al., 2011*; *Yelenik, 2017*). These forests are dominated by two foundational endemic tree species—$N_2$-fixing *Acacia koa* (koa, Fabaceae) and the non-$N_2$-fixing *Metrosideros polymorpha* ('ōhi'a lehua, Myrtaceae)—and provide critical ecosystem services, including habitat for many endangered bird species in Hawai'i (*Paxton et al., 2018*). To restore montane mesic forest habitats on Hawai'i Island, over 390,000

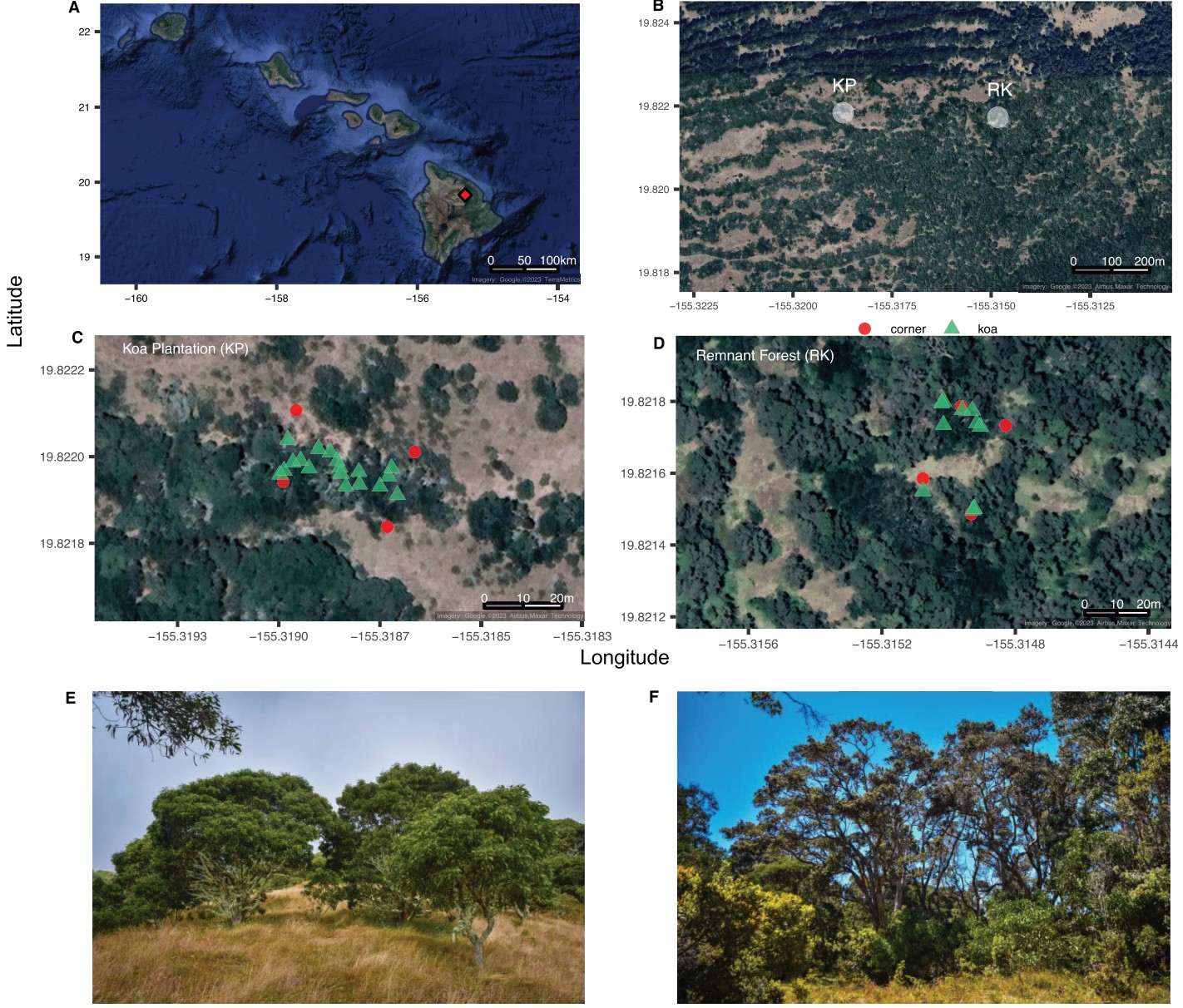

**Figure 1** **The Hawaiian Archipelago and sampling sites.** Site map of (A) the Hawaiian Archipelago highlighting the location of Hakalau Forest National Wildlife Refuge (red diamond) on the island of Hawaiʻi; (B) Koa plantation and remnant forest sampling areas (KP, RK); (C, D) layouts (20 × 35 m), with plot corners in red circles and *Acacia koa* trees in green triangles; (E) images of the koa plantation and (F) perimeter of the remnant forest. Image credit: (A–D) GoogleMaps from R package ggmap, (E, F) L. Kersting.

nursery-grown *A. koa* trees (*i.e.*, outplants) have been planted in the Hakalau Forest National Wildlife Refuge since 1987 (hereafter, Hakalau; Fig. 1). This effort started in the greenhouse, where individual trees (one per pot) were grown in 15-cm-long cone-tainers, reaching a height of 30-cm in <6 months. In the plantation, a bulldozer with a miniblade scoured the soil to remove grass competitors and make plots (1 × 1 m) every 3 m, with rows every 4 m a part. Individual greenhouse-grown koa were planted in holes created by

an auger power planter (the same size as the root plug) with fertilizer added to the auger hole (see *Jeffrey & Horiuchi, 2003*).

*Acacia koa* outplants were introduced to 5,000 acres of fenced, abandoned, previously grazed pastureland in close proximity to remaining forested habitats and endangered bird populations (*Jeffrey & Horiuchi, 2003*; *McDaniel & Ostertag, 2010*). Forest patches are critical in supporting biodiversity (*Wintle et al., 2019*), and patchiness and/or fragmentation can lead to dramatic differences in forest climate that support rates of litter decomposition and nutrient cycling (*Crockatt & Bebber, 2015*). Therefore, the active outplanting of *A. koa* in discrete corridors near more intact forest patches was leveraged to facilitate the passive restoration and expansion of koa and other native plant species recruitments, with the goal of ultimately turning isolated patches into intact, contiguous forest habitat (*Scowcroft & Yeh, 2013*). However, after 30 years the understories of *A. koa* plantations remain grass-dominated, native woody plant recruitment is low, and endangered birds are sparse (*Yelenik, 2017*; *Paxton et al., 2018*). The causes of this stalled forest recovery are uncertain, but they likely include the influence of *A. koa* BNF on soil chemistry, nutrient concentrations, and nutrient cycling (*Scowcroft, Haraguchi & Hue, 2004*). As a $N_2$-fixing tree, *A. koa* produces abundant low C:N leaf litter (high N), which can lead to greater supply of N to the soil surface under *A. koa* relative to *M. polymorpha*, which has high litter C:N (*Scowcroft, 1997*; *Yelenik, Rehm & D'Antonio, 2022*). These differences in litter N can increase rates of decomposition and ultimately affect the return of nutrients to soil N pools (*Baker, Scowcroft & Ewel, 2009*; *Zhou et al., 2018*). Differences in canopy composition and associated litter decomposition can have important implications for forest nutrient cycling and the composition of forest understories. In the plantations, the ability for *A. koa* to increase soil nutrient, coupled with the high densities of this canopy tree in monoculture corridors, may be favoring the growth of exotic nitrophilous grasses from the historic pastures and inhibiting native seedling establishment (*Yelenik, 2017*).

To examine the interactions between BNF, soil and leaf N content as well as C inputs at the ecological community scale, we applied a spatially explicit sampling approach within a mixed *M. polymorpha*/*A. koa* native montane forest (hereafter, the "remnant forest") and an *A. koa* plantation (hereafter, "koa plantation"). The remnant forest is largely dominated by native understory plants, whereas the koa plantation understory is dominated by $C_4$ exotic grasses interspersed with a $C_3$ non-native shrub (*Rubus argutus*). While the land-use history and present-day plant communities of these two forest types differ, they occupy the same climate and parent soil types (http://rainfall.geography.hawaii.edu, https://gis.ctahr. hawaii.edu/SoilAtlas). We used $\delta^{15}N$ and $\delta^{13}C$ values of soil and foliar samples to construct isoscapes and examine the degree of BNF between the two forest types and test the influence of BNF by *A. koa* on $\delta^{15}N$ values in soils and neighboring plants (*Rubus* spp.). We hypothesized that the higher density and more even distribution of *A. koa* in the koa plantation compared to the remnant forest will result in lower $\delta^{15}N$ values in soil and plants from the koa plantations with greater and spatially more homogeneous BNF inputs than the remnant forests. In addition, we expected higher foliar $\delta^{13}C$ values in the koa plantation as a result of greater water demand in the young, dense *A. koa* forests driving
higher WUE$_i$ (*Kagawa et al., 2009*). Introduced C$_4$ grasses are abundant in the understory of the koa plantation but are uncommon in the remnant forest where there are no known native C$_4$ plants (*Yelenik, 2017*). Therefore, we also predicted that soil δ$^{13}$C would be higher in the koa plantation relative to the remnant forest, indicating greater contributions of C$_4$ grasses to soil C pools. The results of this study will clarify the effects of tropical N$_2$-fixing monoculture plantations on the distribution of N and soil nutrient cycling and how these may differ from the target ecosystem for restoration.

# MATERIALS AND METHODS

## Site description and sample collection

Two sections of remnant forest and koa plantation were identified within the Hakalau Forest National Wildlife Refuge on the Island of Hawaiʻi (19°49′12″N, 155°19′22″W) (Fig. 1). Remnant forests are characterized by a mixed canopy of *Acacia koa* (koa) and *Metrosideros polymorpha* (ʻōhiʻa lehua) along with native understory woody plants *Cheirodendron trigynum* (ʻōlapa), *Coprosma rhynchocarpa* (pilo), *Leptecophylla tameiameia* (pukiawe), *Myrsine lessertiana* (kōlea), *Rubus hawaiensis* (ʻākala), and *Vaccinium calycinum* (ʻōhelo). In contrast, the plantations are largely monoculture stands of *A. koa*, with non-native grass understories of *Cenchrus clandestinus* (kikuyu, a C$_4$ grass), along with *Ehrharta stipoides* (C$_3$), with sporadic occurrences of the non-native shrub *Rubus argutus* and ferns.

Research and field collections in Hakalau Forest National Wildlife Refuge were permitted under US Department of the Interior Fish and Wildlife Service Special Use Permit SUP 12516-20015-R. Each sampling area—koa plantation (KP) or remnant forest (RK)—had similar montane forest climates, elevations (~1,800 m), and soils from the same parent material (*Scowcroft et al., 2007*), and were selected due their proximity to transects used in previous studies (*Paxton et al., 2018*). Both the koa plantation and the remnant forest were dominated by *A. koa*, with koa representing 100% of the canopy tree species in the plantation. ʻŌhiʻa was absent in the koa plantation, and only one mature ʻōhiʻa was observed in the remnant forest area. Each 700 m$^2$ sampling area was representative of the vegetation found throughout the remnant and "restored" plantation forests of Hakalau. Samples were collected at a spatial resolution (0–5 m) relevant for detection of BNF and changes in C cycling at the landscape scale in soil, N$_2$-fixing and non-N$_2$-fixing foliage. Individual samples were also collected at a spatial resolution that balanced considerations of spatial autocorrelation and the ability to extrapolate *via* kriging (see methods section below) to the landscape scale (*Hellmann et al., 2011*; *Rascher et al., 2012*).

In each forest plot of 20 × 35 m (Fig. S1), we collected soil samples at 4 or 5-m intervals (*n* = 48 soil samples per forest) using a sterilized (70% ethanol) soil borer that sampled the upper 10 cm of the soil after surface leaf litter was removed. Soil samples were placed into paper envelopes for future processing. Within the remnant forest and the plantation, we collected foliar samples from all mature *A. koa* canopy trees and understory *Rubus* spp. (detailed below). Juvenile *A. koa* have bipinnately-compound true leaves that develop into phyllodes during maturation. Hereafter, we will refer to all foliar samples (*i.e.*, *A. koa* phyllodes, *Rubus* true leaves) as 'leaves'. Grasses were not sampled due to grasses being
rare in the remnant forest. Similar to other plant derived isoscapes (*Hellmann et al., 2011*; *Rascher et al., 2012*), this sampling scheme was designed to capture any signature of $N_2$-fixation from *A. koa* on understory plants as well as the soil environment.

In both forests, all mature *A. koa* larger than 2 m tall were marked (plantation ($n = 18$) and remnant ($n = 10$), 257 and 142 trees per hectare, respectively) and their relative position mapped (Figs. 1 and S1). Three leaves were collected and pooled from each individual *A. koa* tree in the mid-exterior forest canopy using pole pruners at a consistent height (ca. 6 m). The pooled leaf sample for each *A. koa* was placed into a paper envelope until further processing for isotope analyses (detailed below). Due to the distance between individual koa trees, each pooled leaf sample per tree is considered an independent replicate. Each tagged *A. koa* was measured for diameter at breast height (dbh) using a diameter tape, measured at 1.5 m from the ground. When multiple trunks were found on the same tree, dbh was represented as the square root of the sum of squared dbh measurements of individual trunks (*Meeuwig, Cooper & Forestry Sciences Laboratory (Missoula M), 1981*). Canopy area ($m^2$) was measured for each *A. koa* in the forest type using a laser to measure distance from the trunk to the canopy edge; measurements were taken at four bearings (0°, 90°, 180°, 270°). Individual *A. koa* canopy area (canopy $m^2$) was determined using the formula for the area of an ellipse (area = $\pi ab$), where *a* and *b* are the average distances at the major radii (0° and 180°) and minor radii (90° and 270°).

Two Rosaceae species (*Rubus* spp.) were present in the two sampling areas and were used as indicator plants for the influence of BNF on neighboring plants based on their $\delta^{15}N$ values: the native *Rubus hawaiensis* in the understory of the remnant forest ($n = 14$ plants, 200 plants hectare$^{-1}$) and the invasive *Rubus argutus* in the plantation ($n = 28$ plants, 400 plants hectare$^{-1}$). Since the two *Rubus* species tended to separate by forest type, it was not possible to collect both species in the two forests. When a *Rubus* individual was present in a quadrat (20 $m^2$), three leaves were collected from the top of up-to three adult stems and pooled per individual plant and placed in a paper envelope. The spatial location of the *Rubus* samples was standardized to the centroid of the quadrat where they were collected (Fig. S1). Within 6 h of collection, soil and foliar samples were transported to the University of Hawai'i-Hilo Hakalau Forest Biological Field Station and oven dried (60 °C, ca. 5 h) in article envelopes. The dried samples were transported to the BayCEER Laboratory of Isotope Biogeochemistry at the University of Bayreuth for processing.

## Stable isotope analyses

Prior to final analysis, foliar and soil samples were again oven dried (105 °C) to a constant weight (*Gebauer & Schulze, 1991*) and homogenized to a powder using a ball mill (Retsch Schwingmühle MM2, Haan, Germany). While many studies use 60 °C for foliar samples, extensive internal laboratory testing of drying procedures and temperatures (60°, 80°, 105 °C) have shown negligible drying effects on foliar and soil isotope values (within ± 0.2‰ measurement error), while drying at higher temperatures ensures complete sample desiccation. Where present, root material was removed from soil samples prior to homogenization. A subset of each sample was placed in a tin capsule and weighed (foliar (2.8–5.5 mg), soil (3.5–4.5 mg)). A standard material (acetanilide (0.4–1.5 mg)) with

known C and N stable isotope abundance and concentration was also analyzed for quality control and calculation of C and N concentrations in our samples. Standard materials were run at least six times within each batch of 50 samples following protocols previously described (*Gebauer & Schulze, 1991*). Molar concentrations of C and N and their mass percent (%) ratio (C:N), as well as natural abundance isotope values ($\delta^{13}$C, $\delta^{15}$N), were measured using an elemental analyzer-isotope ratio mass spectrometer (EA-IRMS) coupling, combining an elemental analyzer (Carlo Erba Instruments 1108, Milano, Italy) with a continuous flow isotope ratio mass spectrometer (delta S, Finnigan MAT, Bremen, Germany) *via* a ConFlo III open-split interface (Thermo Fisher Scientific, Bremen, Germany). Natural abundance isotopic values were reported in delta values ($\delta$) using permil (‰) notation relative to standard materials (Vienna-Peedee Belemnite (V-PDB) and atmospheric $N_2$ standards (air) for C and N, respectively). Reproducibility of isotope abundance measurements was always within ± 0.2‰. All standard gases (Riessner, Lichtenfels, Germany) were calibrated *vs.* the international standards using reference substances provided by the International Atomic Energy Agency, Vienna, Austria (CH-3, CH-6 and NBS-18 for the carbon isotopes and N-1 and N-2 for the N isotopes).

## Isoscape kriging

Spatially explicit sampling of soil, *A. koa*, and indicator plant taxa (*Rubus* spp.) allowed for the generation of maps of $\delta^{15}$N and $\delta^{13}$C values across the two forest types (*i.e.*, isoscapes). The spatial coordinates for each sample were determined by its location within the sampling grid (*i.e.*, quadrat corners (soil), centroid (*Rubus* spp.), or relative position (*A. koa*)), which was used to generate a spatial grid and points for interpolation using the functions *coordinates* and *spsample* in the R package "sp" (*Pebesma & Bivand, 2005*; *Bivand, Pebesma & Gómez-Rubio, 2013*). To visualize the stable isotope landscape, we used kriging: a common statistical method that uses variograms to calculate the spatial autocorrelation between points and distance and interpolate isotope values across a continuous surface (*Bowen, 2010*). We generated $\delta^{15}$N and $\delta^{13}$C isoscapes for soil-only samples and a $\delta^{15}$N isoscape for foliar-only samples. A foliar $\delta^{13}$C isoscape was not generated since samples were limited to $C_3$ plants of similar $\delta^{13}$C values. Data interpolation was performed using ordinary kriging to a continuous surface, with variogram models generated by the function *autoKrige* in the R package "automap" (*Hiemstra et al., 2009*). *AutoKrige* performs iterative model selection fitting variogram models to the collected data (the experimental variograms). Best-fit variogram models (*i.e.*, those with lowest residual sum of squares) have four main model parameters: *the nugget effect*, the estimate of experimental error inherent in measurements, sampling design, and environmental variability; *the sill*, the spatial pattern intensity and the semivariance asymptote; *the spatial range*, the distance or lag of sample correlation where samples are influenced by same underlying process; and *the sserr*, the sum of squares between experimental and fitted variogram model (*Fortin, Dale & Ver Hoef, 2012*). These model parameters describe the autocorrelation between the semivariance (y-axis) and distance or lag parameter (x-axis). With distance between samples, semivariance increases and correlation decreases; therefore, model saturation (*i.e.*, *sill*) indicates a point where

semivariance no longer increases with increasing distance (*i.e.*, *range*) and points no longer show autocorrelation. We used variogram models and their predicted values ($\delta^{15}N_{predicted}$ and $\delta^{13}C_{predicted}$) to generate isoscapes using the function *spplot* in the R package "sp". Where model variograms showed poor fit to experimental variogram data, we used *autoKrige* to run iterations of user-defined, fixed values for variogram parameters (*nugget*, *range*, *sill*). Predicted values from model variograms were then compared between the two forest types over a continuous surface with statistical inferences made on the distributions of predicted values using non-parametric tests.

## Statistical analyses

Data normality and heteroscedasticity were inferred from graphical inspection of residuals and quantile-quantile plots. Where data failed to meet assumptions of parametric models, non-parametric tests were used. Differences in *A. koa* dbh and canopy area between forest types (koa plantation or remnant forest) were analyzed using Mann-Whitney *U*-tests using the function *mwu* in the package "sjstats" (*Lüdecke, 2021*). Isotope data ($\delta^{15}N$, $\delta^{13}C$), total C and total N concentration (mmol/g dry weight (gdw)), and mass % C:N were analyzed using a two-way linear model with sample type (*A. koa*, *Rubus* spp., soil) and forest type (remnant forest, koa plantation) as fixed effects. Analysis of variance tables were generated with Type-II sum of squares in the *car* package (*Fox & Weisberg, 2019*). *Post-hoc* slice-tests were used in *a priori* contrasts to test for significant differences among sample types within a forest (*A. koa vs. Rubus* spp.) and between forest types using the *emmeans* package (*Lenth, 2022*). Differences between the $\delta^{15}N$ and $\delta^{13}C$ isoscapes in the remnant forest and koa plantation were assessed visually, through Mann-Whitney *U*-tests using density plots of interpolated/predicted isotope values ($\delta^{15}N$ or $\delta^{13}C$), and the characteristics of the model variogram.

While we recognize our study is limited to two areas ($20 \times 35$ m) each within a different forest type (koa plantation and remnant forest), the utility of an isoscape approach is that it allows for spatial interpolation so that biogeochemical patterns at community and ecosystem levels can be quantified. Experimental designs such as ours, where a single large area representative of the community types to be compared, are common and accepted approaches among isoscape studies assuming that sampling is sufficient to capture natural variation in isotope values within the community of interest (*Bowen, 2010*; *Hellmann et al., 2011*; *Rascher et al., 2012*). Here, the unit of replication is individual sample points used to generate isoscape maps (plants ($n = 24–36$), soil ($n = 48$)), with statistical relationships determined using linear models. All data analyses were performed in R (version 4.2.1) (*R Core Team, 2022*). Archived data and code for analyses can be found at GitHub (https://github.com/cbwall/Hakalau) and are published at Zenodo (*Wall, 2023*).

# RESULTS

## Vegetation structure

Differences between *A. koa* trees in the two forest types were observed, notably dbh was greater in the koa plantation ($p = 0.010$) (Fig. S2A), but the *A. koa* canopy area was not different between the two forests ($p = 0.666$) (Fig. S2B, Table S1). In the remnant forest,

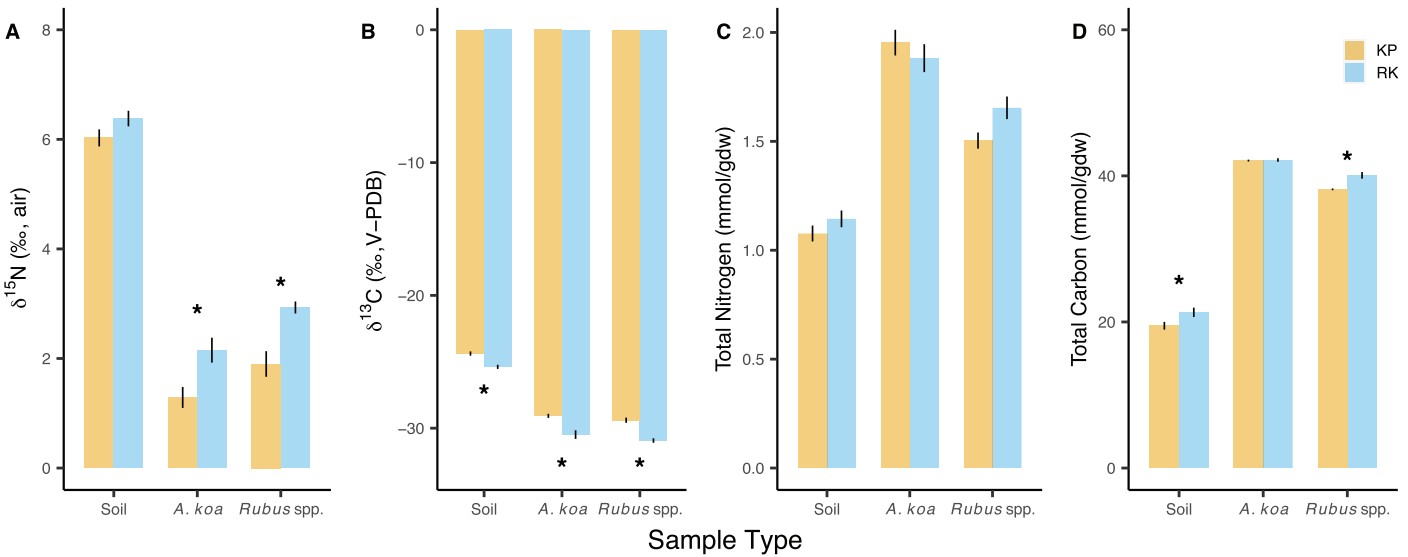

**Figure 2 Soil and foliar isotope values and elemental concentrations.** Soil and foliar (*Acacia koa, Rubus* spp.) (A) $\delta^{15}$N, (B) $\delta^{13}$C, (C) total nitrogen, and (D) total carbon in koa plantation (KP) and remnant (RK) Hakalau forests. Asterisks (*) indicate significant differences between forests within a sample type ($p < 0.05$). Values are mean ± SE, $n$ = 48 (soil), 18 and 10 (*Acacia koa*), 28 and 14 (*Rubus* spp.) in KP and RK forests, respectively.

80% of *A. koa* were single-stemmed, whereas in the koa plantation multi-stemmed *A. koa* dominated with only 20% of trees being single-stemmed. The total basal area of mature *A. koa* cover in the koa plantation and remnant forest (18 and 10 trees, respectively) was 283 and 258 m², covering ~40% of the 700 m² sampling areas in each forest.

## Soil and foliar isotope values

$\delta^{15}$N values differed according to sample type ($p < 0.001$) and forest ($p < 0.001$), with overall lower $\delta^{15}$N values in the koa plantation (Table S2). $\delta^{15}$N values of soils were not different among the forests ($p = 0.085$). *Acacia koa* and *Rubus* spp. $\delta^{15}$N values were both significantly closer to zero (less $^{15}$N-enriched by ~1‰) in the koa plantation compared to remnant forest samples (*post-hoc*: $p \le 0.029$) (Fig. 2A). As expected, $\delta^{15}$N values of the $N_2$-fixing *A. koa* were significantly lower compared to *Rubus* spp. in both the koa plantation (*post-hoc*: $p = 0.020$) and remnant forest (*post-hoc*: $p = 0.029$). Overall, there was a greater range and standard deviation in $\delta^{15}$N values in the remnant forest *A. koa* (range: 2.19‰, SD: 1.03‰) and *Rubus* spp. (range: 3.54‰, SD: 0.89‰) compared to plantation *A. koa* (range: 0.85‰, SD: 0.65‰) and *Rubus* spp. (and 2.55‰, SD: 0.75‰). This larger range and variance in plant $\delta^{15}$N values is consistent with heterogeneous N sources in the remnant forest, which is being expressed in the leaves of canopy $N_2$-fixing trees and understory plants. Soil $\delta^{15}$N values were ~3‰ higher than those in foliar samples, but similar among the koa plantation and the remnant forest (*post-hoc*: $p = 0.085$) (Fig. 2A).

$\delta^{13}$C values were different among sample types ($p < 0.001$) and between forests ($p < 0.001$), with significantly lower $\delta^{13}$C values in all sample types from the remnant forest relative to those from the koa plantation (Fig. 2B; Table S2). However, when compared within the same forest, the $\delta^{13}$C values of *A. koa* and *Rubus* spp. were not different from

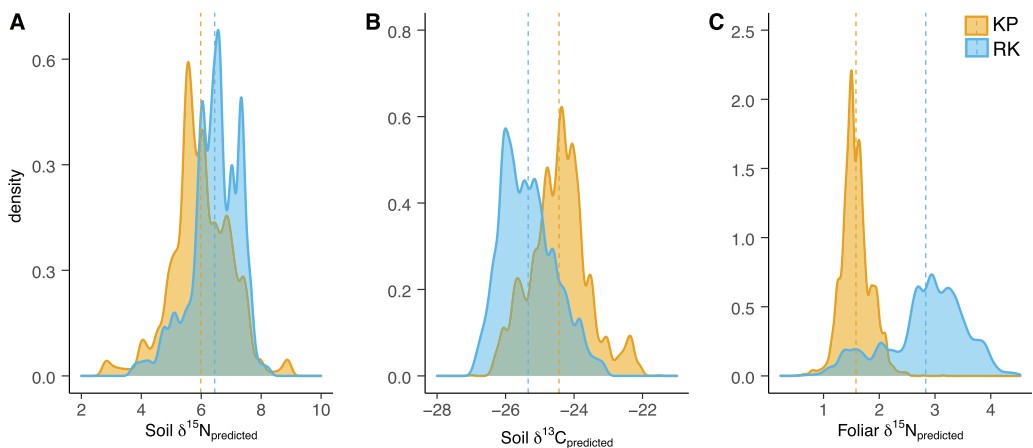

**Figure 3 Soil and foliar isoscape density plots.** (A, B) Density plot of soil $\delta^{15}N_{predicted}$ (*left*), soil $\delta^{13}C_{predicted}$ (*middle*), and (C) foliar $\delta^{15}N_{predicted}$ samples (*Acacia koa* and *Rubus* spp., *right*) from isoscapes in the koa plantation (KP) and remnant (RK) Hakalau forests. Dashed lines indicate mean values in each forest.

each other in the remnant forest (*post-hoc*: $p = 0.057$) or the koa plantation (*post-hoc*: $p = 0.281$).

## Isoscapes and model variograms

Variograms provided insights into the spatial correlation within isoscapes and the semivariance-by-distance relationship. $\delta^{15}N$ and $\delta^{13}C$ isoscape variograms for both soil and leaves showed equivalent ranges in spatial autocorrelation in both forests (~3–5 m) (Table S3). These results indicate similar relationships between semivariance-by-distance decay (*i.e.*, less autocorrelation with increasing distance) in the koa plantation and the remnant forest at the scale of our experimental sampling. However, these estimates should be interpreted with caution since (1) the minimum sampling distance between samples for soils was 4 m and (2) in several modeled variograms the increased *nugget* values coupled with low *sill* values may indicate greater influence of random variation. Nevertheless, we found significantly different distributions of $\delta^{15}N$ and $\delta^{13}C$ isotope values between forest types for both soil and foliar isoscapes ($p < 0.001$) (Figs. 3–5; Table S1).

In both soil and foliar $\delta^{15}N$ isoscapes, we observed lower $\delta^{15}N_{predicted}$ values in the koa plantation ($p < 0.001$) (Table S1), with mean $\delta^{15}N$ values for all interpolated data points lower by ~0.5‰ (soil isoscape) and 1.2‰ (foliar isoscape) in the plantation relative to the remnant forest (Figs. 3A–3C). Both forests showed hot spots of low $\delta^{15}N$ values (<4‰ soil Figs. 4A and 4B, <2‰ foliar Figs. 5A and 5B) that tended to be in areas proximate to *A. koa* or sampling points representing *A. koa* leaves (and to a lesser extent *Rubus* spp.) (Figs. 4 and 5). However, low $\delta^{15}N$ areas were most pronounced in the koa plantation, where *A. koa* density was greater than the remnant forest (18 KP *vs.* 10 RK mature trees), particularly through the middle of the koa plantation where *A. koa* trees are numerous and planted in parallel rows. Similarly, in the $\delta^{15}N$ soil-isoscape, the koa plantation had more uniform low $\delta^{15}N$ areas within this central *A. koa* corridor (Fig. 4A). The influence of

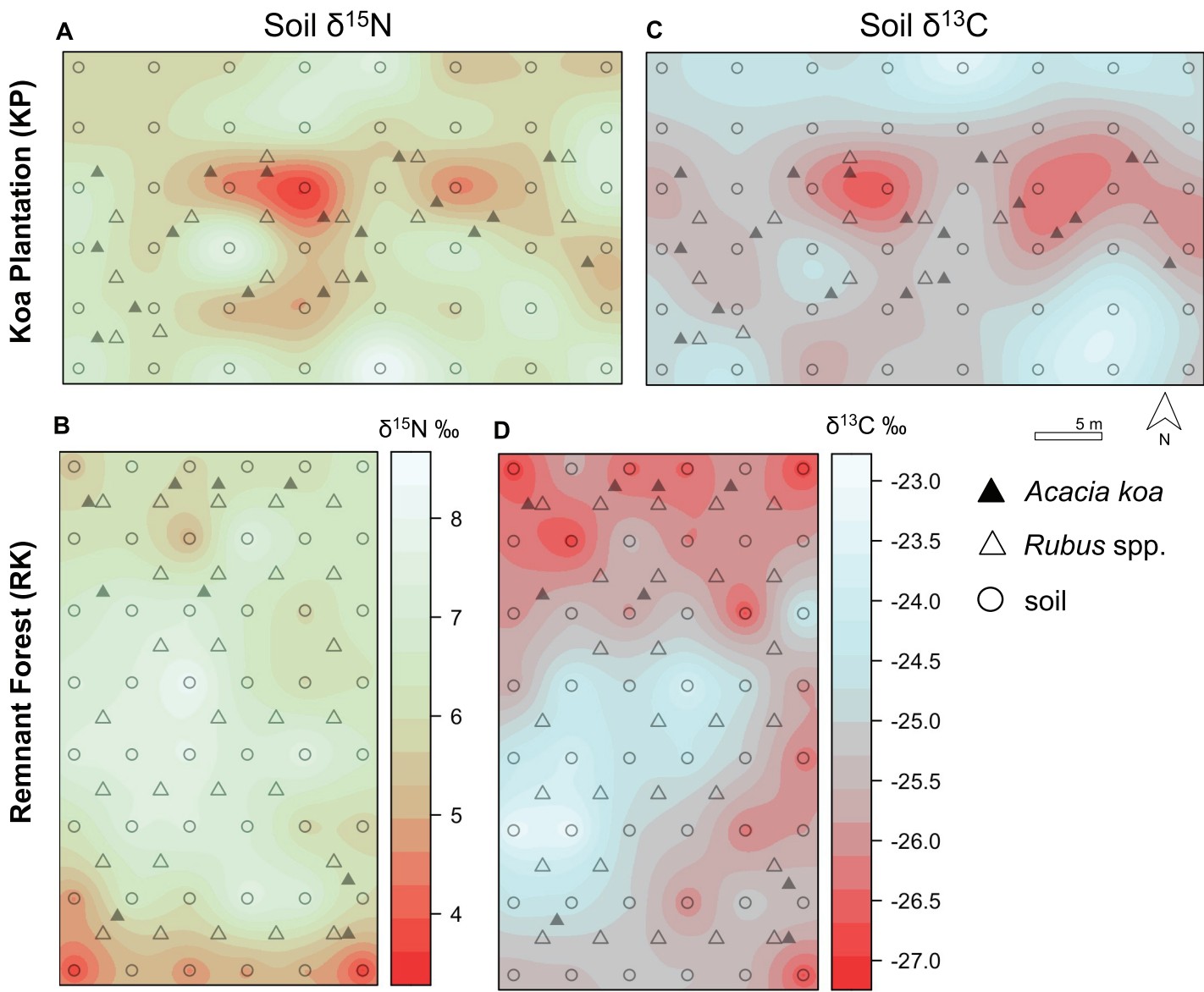

**Figure 4 Soil δ¹⁵N and δ¹³C isoscapes.** (A, B) Soil δ¹⁵N and (C, D) δ¹³C isoscapes for Hakalau koa plantation (KP, top) and remnant forests (RK, bottom). Color bar represents δ¹⁵N and δ¹³C values. Symbols represent locations where soil (open circles), *Acacia koa* (filled triangles), and *Rubus* spp. (open triangles) samples were collected; only soil samples were used in spatial interpolation.

*A. koa* on soils in the remnant forest was more variable, with a localized region of low δ¹⁵N values in the southern portion of the forest plot (Fig. 4B).

δ¹³C isoscapes of soil samples showed lower δ¹³C values throughout the remnant forest relative to the koa plantation (Figs. 4C and 4D), with the distribution of δ¹³C$_{\text{predicted}}$ values for soil samples (Fig. 3B) being lower in the remnant forest ($p < 0.001$) (Table S1). These differences represent ~1.5‰ lower mean δ¹³C values for interpolated values in the remnant forest compared to the plantation. Hotspots of low δ¹³C values in both δ¹³C soil isoscapes corresponded to areas within the *A. koa* corridor in the plantation and areas where juvenile *A. koa* were located (Figs. 4C, 4D and S1).

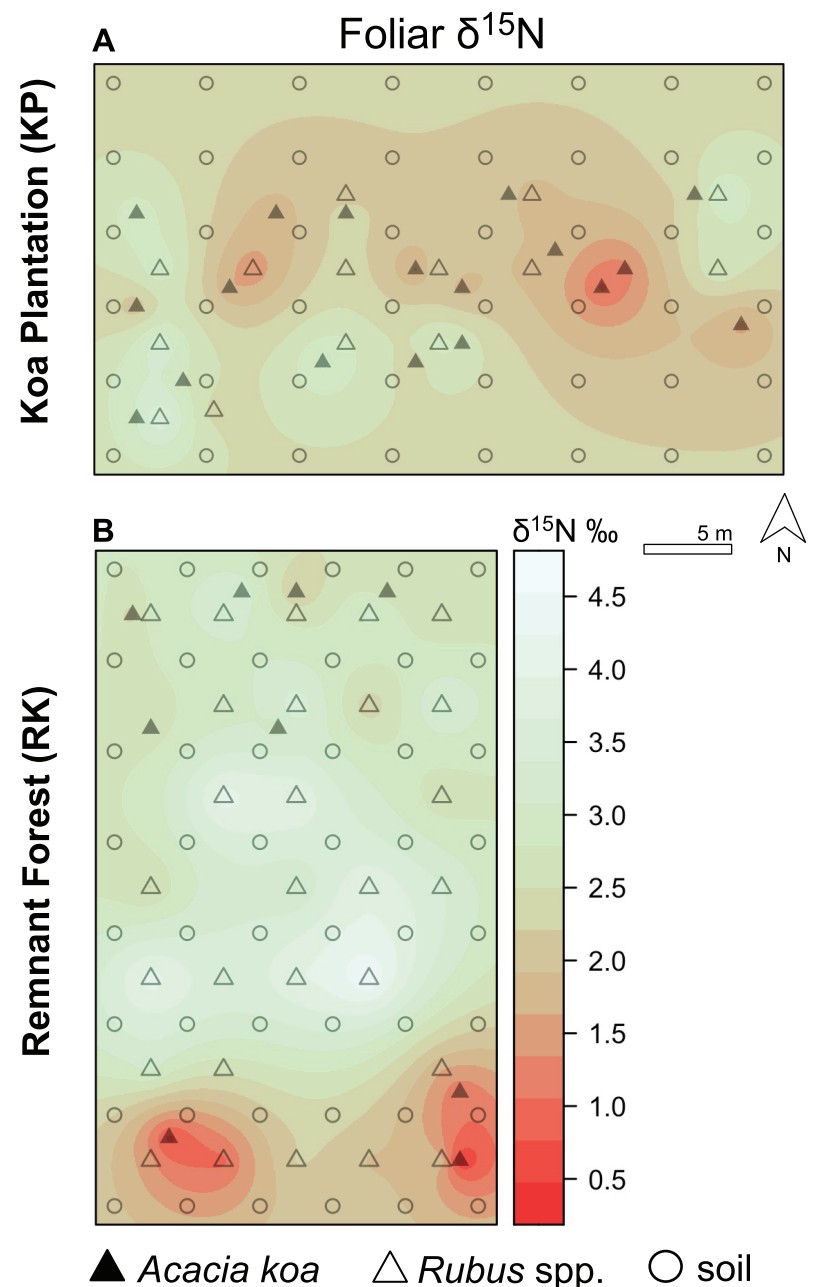

**Figure 5  Foliar δ¹⁵N isoscapes.** (A) Foliar $\delta^{15}N$ for Hakalau koa plantation (KP, top) and (B) remnant forests (RK, bottom). Color bar represents $\delta^{15}N$ values. Symbols represent locations where soil (open circles), *Acacia koa* (filled triangles), and *Rubus* spp. (open triangles) samples were collected; only foliar samples were used in spatial interpolation.               

## Nitrogen and carbon concentration patterns

N concentration (mmol/gdw) differed according to sample types ($p < 0.001$) (Fig. 2C), being highest in *A. koa*, followed by *Rubus* spp., and lowest in soil samples ($p < 0.001$) but did not differ between forests ($p = 0.107$) (Table S2). Total C concentrations (mmol/gdw) differed among sample types ($p < 0.001$) and were overall lower in the koa plantation

($p$ = 0.003), driven by small differences in C concentrations of soil and *Rubus* sp. (<2%) (Fig. 2D). N and C concentrations were greater in *A. koa* compared to *Rubus* spp. in the plantation (*post-hoc*: $p < 0.001$) and the remnant forest (*post-hoc*: $p \leq 0.022$). Molar C:N values differed by sample types ($p < 0.001$), but not forests ($p = 0.621$), and were lowest in soil (16), followed by *A. koa* (19), and *Rubus* spp. (21) (pooled means ± 0.3 SE, $n = 14–96$).

## DISCUSSION

Assessments of biogeochemical landscapes can provide insight into key ecosystem properties, plant-soil feedback, and plant-plant interactions. As the degree of BNF increases, there is a well-documented decrease in foliar $\delta^{15}N$ in $N_2$-fixing plant species (*Craine et al., 2015*). Differences in BNF between ecological communities often have corresponding effects on other ecosystem properties such as soil and neighboring non-$N_2$-fixing plant $\delta^{15}N$ values and nutrient status, as well as rates of photosynthesis and nutrient cycling. In addition to $\delta^{15}N$ analyses, $\delta^{13}C$ values coupled with information on C and N concentrations can aid in inferences about each of these ecosystem-level processes occurring at different spatial scales (*Moyer-Henry et al., 2006*; *Rascher et al., 2012*; *Hoogmoed et al., 2014*; *Craine et al., 2015*). The use of large "super-plots" in discrete habitats, which are sampled at small spatial scales and/or grids to perform spatial interpolation, is common in isoscape experimental designs (*Rascher et al., 2012*; *Hellmann et al., 2016*). Earlier studies sampled *A. koa* across elevation and rainfall gradients on Hawai'i Island, mapping $\delta^{15}N$ and $\delta^{13}C$ values to determine environmental impacts on plant performance and suitability of sites for forest restoration (*Lawson & Pike, 2017*). We find that spatially-explicit sampling at smaller spatial scales is equally useful in identifying difference is plant-soil-microbe interactions and the influence of BNF. Here, using $\delta^{15}N$ isoscapes to determine differences in BNF between two forest patches that differ in their land use histories and management, we found strong evidence for greater, and more homogeneous contribution of newly-fixed N in the soil and plants of the afforested koa plantation compared to the remnant native forest. While these forests plots represent but a subset of the plantation and remnant forest habitats, each plot covered a significant area (700 m$^2$) and was specifically chosen because the vegetation was representative of each unique forest type, while the abiotic conditions were similar (climate, slope, aspect, soil parent material). Therefore, the results of our study should be considered representative of forest community-level dynamics. However, increasing plot-level replication in future studies may allow for greater biogeochemical heterogeneity to be quantified within and among forest types (plantations and remnant forests) and the influence of environmental factors (such as elevation, rainfall, volcanic organic gases) on BNF and forest restoration.

In the koa plantation, $\delta^{15}N$ soil and foliar isoscapes revealed a more evenly distributed signature of low $\delta^{15}N$ values in the plantation relative to the remnant forest, indicative of a greater contribution of BNF in the koa plantation that could be related to the demography and density of *A. koa* (Figs. 4 and 5). The finding that both *A. koa* and understory *Rubus* spp. also had lower $\delta^{15}N$ values in the koa plantation relative to the remnant forest (Fig. 2) further suggests greater BNF by *A. koa* and a greater contribution of newly-fixed N to

neighboring plants. While soil $\delta^{15}N$ values did not statistically differ significantly between forest types, the $\delta^{15}N$ soil isoscape showed overall lower $\delta^{15}N$ values in the koa plantation compared to the remnant forest (Fig. 4). The marginal differences in soil $\delta^{15}N$ between the two forest types may be due in part to differences in water availability between forest types (*Austin & Vitousek, 1998*), but further efforts are needed to determine if the soil water status differs among these forests (*Bothwell et al., 2014*). In both forests, however, the high soil $\delta^{15}N$ values indicate significant losses of N relative to the size of the nitrogen pool, possibly from rapid N turnover and fractionation or leaching (*Natelhoffer & Fry, 1988*; *Austin & Vitousek, 1998*; *Burnett et al., 2022*). Nevertheless, our suite of biogeochemical metrics ($\delta^{15}N$, $\delta^{13}C$, N and C concentrations) provide new insights on plant-soil-water relations, nutrient and C cycling, and offer clues as to why the plantations continue to foster a non-native grass dominated state even after three decades post reforestation (*Yelenik, 2017*; *Yelenik et al., 2021*).

In addition to higher density of *A. koa* in the plantation, the demographics of *A. koa* outplants and saplings in the plantation (<30 y old) compared to the remnant forest may be influencing C and N cycling in this system. Nodule biomass and $N_2$-fixation rates are life-stage dependent, with younger stands (6 y) exhibiting an order of magnitude higher nodule biomass and $N_2$-fixation rates than older stands (20 y) (*Pearson & Vitousek, 2001*). The growth rate of *A. koa* across the Hawaiian Islands is variable, with dbh-based growth rates of 10–15 mm/y in sunlit areas where the crown is exposed and 6–7 mm/y estimated across a range of sites in the Hawai'i Department of Forestry and Wildlife long-term forest plots (*Baker, Scowcroft & Ewel, 2009*). In remnant forests of Hakalau, growth rates of 4 mm/y for *A. koa* have been observed (*Hart, 2010*), but no dbh-age estimates exist for the koa plantation. If we assume *A. koa* in the sunlit plantation have growth rates similar to other sunlit areas across Hawai'i (~12 mm/y) and *A. koa* in the remnant forest growth rates are 4 mm/y, we estimate mean (±SE) ages of trees from the plantation to be significantly younger ($p = 0.049$, Table S1) than the remnant forest (30 ± 2 KP and 55 ± 12 years RK) (Fig. S3), with two trees from the remnant forest being >110 years old. These estimates agree with the known age of the planting of the plantation (~1990) and support the hypothesis that forest demographics may be important to affecting the degree of $N_2$-fixation in *A. koa* and its contribution to plants and soils of Hakalau.

In our study system, the high densities *A. koa* in the plantation, along with immature *A. koa* recruits in thickets adjacent to planted trees, may be contributing to the higher rates of $N_2$-fixation we detected based on lower $\delta^{15}N$ values (Figs. 2A, 4 and 5). The high density of *A. koa* has the potential to contribute more leaf litter of lower C:N to the forest floor relative to the mixed-canopy remnant forest. The greater number of multi-stemmed *A. koa* in the plantation (80% in KP *vs.* 20% RK)—which may be due to the spacing of outplants and the lack of competition in the plantation—also emphasizes the differing growth patterns in the outplanted *A. koa* relative to naturally recruited trees in remnant forests. These conditions may shape litterfall to the forest floor, which can be dynamic in both abundance and nutrient concentrations and reflective of shifts in plant community composition (*Lanuza et al., 2018*). Taken together, we suggest the greater densities, faster growth rates, and younger demographics of *A. koa* in the koa plantation provide a context

for low C:N litterfall (and greater N contributions) in the early stages of reforestation to lead to persistent changes in nutrient cycling.

Despite a strong signal of significantly higher BNF in the plantation (Figs. 2A, 4A and 5A), we found no statistically significant differences in foliar or soil N concentrations (Figs. 2C and 2D). While this result is puzzling, one likely explanation is that the plantation may have relatively more well drained soils, and plantation *A. koa* may have higher water demands compared to remnant forests (*Meinzer, Fownes & Harrington, 1996*; *Brauman, Freyberg & Daily, 2015*). We note *A. koa* foliar $\delta^{15}N$ values reported here are high relative to other studies of *A. koa* (*Burnett et al., 2022*; *Lawson & Pike, 2017*); however, these values are in-line with $N_2$-fixing plants from dry-forests and grasslands (*Heaton, 1987*), suggesting water limitations may be influencing *A. koa* $\delta^{15}N$ in these forests. Even though we did not measure water holding capacity of each soil type, the significantly higher foliar $\delta^{13}C$ values in the plantation—coupled with the lack of differences in soil and foliar N despite evidence of higher BNF—is indicative of water being an important factor limiting *A. koa* productivity and affecting soil N properties (*Ares & Fownes, 1999*; *Burnett et al., 2022*).

Assuming water is more limited in the plantation, higher $\delta^{13}C$ foliar values among *A. koa* and *Rubus* sp. individuals in the plantation may indicate higher rates of photosynthesis (reduced $^{13}CO_2$ discrimination) and/or higher $WUE_i$ relative to the remnant forest (*Farquhar, Ehleringer & Hubick, 1989*; *Cernusak et al., 2013*). Considering the similarity in soil and foliar N in the two forests, we expect the differences in $\delta^{13}C$ values are more likely to relate to $WUE_i$ and not increased rates of photosynthesis. In support of this, (*Ares & Fownes, 1999*) found *A. koa* foliar $\delta^{13}C$ values increased across a gradient of decreasing rainfall, and $WUE_i$ increased in greenhouse grown seedlings experiencing drought-stress. Furthermore, it has been documented that remnant native Hawaiian forests conserve more water (*Kagawa et al., 2009*) and contribute more to aquifer recharge compared to plantation forests (*Brauman, Freyberg & Daily, 2015*). More stable water availability and less draw down of water resources in the remnant forest is consistent with patterns of greater stomatal conductance, higher ci/ca, and lower $WUE_i$ (*Farquhar, Ehleringer & Hubick, 1989*). Therefore, lower foliar $\delta^{13}C$ values in *A. koa* and *Rubus* sp. in the remnant forest may be an effect of less water demand in native/remnant forests, better soil water holding capacity, or both (Fig. 2). This finding is important, as the greater abundance of $C_4$ grasses in the understory of the plantation (*Yelenik, 2017*) may relate to the greater competitive ability of these grasses for water resources compared to native understory plants.

Photosynthetic fractionation is less in $C_4$ than $C_3$ plants, resulting in $^{13}C$-enrichment and higher leaf $\delta^{13}C$ values ranging from −10‰ to −15‰ compared to $C_3$ plants (−21‰ to −30‰) (*Farquhar, Ehleringer & Hubick, 1989*; *Ehleringer, Buchmann & Flanagan, 2000*). In the koa plantation soils, a clear pattern emerges of higher soil $\delta^{13}C$ values, consistent with a greater contribution of $C_4$-derived C to soils (*Staddon, 2004*). This finding is also supported by the soil $\delta^{13}C$ isoscape (Fig. 4), where we observed increasing soil $\delta^{13}C$ values in areas where grasses were dominant. N availability also influences rates of decomposition and has a direct influence on soil C pools (*Averill & Waring, 2018*); therefore, greater BNF

in the plantation could accelerate soil C turnover, which may result in soils enriched in $^{13}$C (*Choi et al., 2005*). Accordingly, future studies should examine isotope values in understory grasses and how low C:N leaf litter, such as *A. koa*, is assimilated into soil and understory plant biomass, especially in the nitrophilous kikuyu grass (*Cenchrus clandestinus*). Follow up studies are on-going to determine rates of soil nutrient cycling in Hakalau, especially in order to disentangle how soil C, N, and leaf litter C:N influence soil $\delta^{13}$C values.

## CONCLUSION

Recuperating biogeochemical properties in secondary tropical forests is vital to supporting plant growth and post-disturbance recovery (*Sullivan et al., 2019*). In some cases $N_2$-fixing plants may therefore be useful in facilitating nutrient and plant community recovery (*Chaer et al., 2011*). However, our data suggest that monoculture plantations of a native $N_2$-fixing tree can lead to an increase in BNF and a more homogeneous distribution of fixed N in plants and soils that is unlike the distribution of N in a primary forest—the target habitat for restoration (Figs. 4 and 5). If this "plantation-effect" promotes undesirable species such as the non-native grasses in our plantation site, then creating a homogeneous area of high BNF may hinder restoration goals. Therefore, restoration efforts should consider how the abundance and density of $N_2$-fixing trees may influence plant-plant and plant-soil interactions and the potential for restoration to produce ecosystem states that differ from reference ecosystem counterparts. Considering the wide application of $N_2$-fixing trees in tropical forest restoration, both in Hawai'i and abroad, there remains a need to better understand how BNF by canopy trees varies within-and-among forest areas, across environmental conditions and habitat types, and its role in affecting nutrient cycling and forest communities. Addressing this uncertainty will support effective restoration strategic planning (*i.e.*, outplanting density, multiple species planting) and management goals (*i.e.*, habitat restoration, seedling and avian recruitment). Based on our findings in Hakalau, we suggest future restoration efforts might include a greater diversity of $N_2$-fixing and non-$N_2$-fixing canopy tree species to generate greater variation in understory and soil conditions more typical of remnant or intact forests.

## ACKNOWLEDGEMENTS

We acknowledge field assistance from students from Bayreuth University (Interdisciplinary Field Course on Ecological Interactions within the graduate classes Biodiversity and Ecology and Molecular Ecology): A. Augenstein, C. Greiner, L. Kersting, L. Pelzer, K. Wobedo and F. Zahn. We acknowledge G. Runte and J. Stallman for field assistant and logistical support; C. Tiroch and C. Bauer for technical assistance in isotope ratio mass spectrometry; M. Kantar, S. Yelenik and P. Hart for discussions on Hakalau and kriging.

### Funding

This work was supported by the National Science Foundation (award numbers 1556856, 1557177, 2124922) and a Junior Researcher International Fellowship from the University of Bayreuth to Nicole A. Hynson. The funders had no role in study design, data collection and analysis, decision to publish, or preparation of the manuscript.

### Grant Disclosures

The following grant information was disclosed by the authors:
National Science Foundation: 1556856, 1557177, 2124922.
Junior Researcher International Fellowship from the University of Bayreuth to Nicole A. Hynson.

### Competing Interests

The authors declare that they have no competing interests.

### Author Contributions

- Christopher B. Wall conceived and designed the experiments, performed the experiments, analyzed the data, prepared figures and/or tables, authored or reviewed drafts of the article, and approved the final draft.
- Sean O. I. Swift conceived and designed the experiments, performed the experiments, analyzed the data, prepared figures and/or tables, authored or reviewed drafts of the article, and approved the final draft.
- Carla M. D'Antonio conceived and designed the experiments, authored or reviewed drafts of the article, and approved the final draft.
- Gerhard Gebauer conceived and designed the experiments, performed the experiments, analyzed the data, authored or reviewed drafts of the article, and approved the final draft.
- Nicole A. Hynson conceived and designed the experiments, performed the experiments, analyzed the data, authored or reviewed drafts of the article, and approved the final draft.

### Field Study Permissions

The following information was supplied relating to field study approvals (*i.e.*, approving body and any reference numbers):
US Department of Interior Fish and Wildlife Service permit ID 12516-20015-R.

### Data Availability

All data and code are available at GitHub and Zenodo: https://github.com/cbwall/Hakalau-Isoscapes.
Chris Wall. (2023). cbwall/Hakalau-Isoscapes: pub.ver (pub.ver). Zenodo. https://doi.org/10.5281/zenodo.7901603.

## Supplemental Information

Supplemental information for this article can be found online at http://dx.doi.org/10.7717/peerj.15468#supplemental-information.

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
