# Peer review of "Isoscapes of remnant and restored Hawaiian montane forests reveal differences in biological nitrogen fixation and carbon inputs"

_PeerJ, doi:10.7717/peerj.15468_

## Round 0.1 · original submission · Minor Revisions

The reviewers found the manuscript to be of interest and within the scope of the journal. They found the experimental design to be generally appropriate and the interpretations of the results valid, though they did express concerns about n=1 and they offer a number of specific suggestions to improve clarity. Please address each reviewer comment/suggestion in your revisions.

Reviewer 1 ·

Basic reporting

The ms does a good job introducing the restoration issues and giving an overview of the data, which is further detailed in archival resources. The nature of the comparisons seems to involve interpolated data points rather than simpler t tests among pooled means of measured data. It might be wise to add some of these simpler tests to strengthen comparisons where they are critically important; so rely on interpolated data for general trends, measured data for important emergent results.

Experimental design

The comparison between just 2 sites is always somewhat iffy, with N = 1 vs. N = 1. However, the multiple sampling “isoscape” points within the two sites allows a differentiated view of patchiness and homogeneity issues that are usually missed in wider forest surveys (e.g. in the nice Austin and Vitousek 1998 study cited in the text). Conclusions about patchiness are important, and part of the story of forest evolution. In this case, the choice of the intensive within-plot replication was satisfying and informative, at least to this reviewer. One suggestion is that the authors add a few sentences in the Introduction about the importance of patchiness in forests, as gaps and for biodiversity?

Validity of the findings

The study deals with about 400,000 trees planted over decades in an area comprising about 20 km2 of new forest. This is a massive project! It is really good to have this 30 year interim report about how things are going. Two points that might need a bit of elaboration are reasons why the plantation trees might have multiple stems (stems keep failing in old poor soil, so trees develop new ones?), and how long forests really need before they “recover” (is it 50 years to 50%....). Also missing is a perspective on what might be done to set right these forests that are depicted as “stalled” in bad trajectories (lines 79 and 137).

Additional comments

The subject is a fascinating one, how to restore a forest for biogeochemistry and bird habitat. An emergent theme was that water might become the critical issue in understanding this restoration. Authors develop this point in the Discussion fairly convincingly. I looked at a detailed map in Google Maps and found that the study sites are near the Mauna Kea saddle where rainfall starts to taper off, with the remnant forest a little (about 500m) downslope and closer to the start of the Hakalau stream. This would be consistent with the “more available water at the RK” site theme that the authors develop. Also, there are fairly high absolute 15N values there averaging 2.9o/oo; foliar values about 2o/oo or so start to be in the range characterizing dry forests and grasslands (see older pre-2000 references about high 15N from South Africa, especially with Heaton as an author; e.g Nature 322, 822-823; 1986). These points about water seem the really new parts of this ms, and for this reason, I strongly support publication.

Specific points:
a. 3 and 4 in C3 and C4 should be subscripts, so C3 and C4
b. Line 138. Rates of decomposition; expound a bit on how this can influence forest structure; do mature forests have slow rates? Is the issue what favors nutrients to plants vs. sequestered by microbes in soils? Explain for the general reader what the restoration issue might be.
c. Line 326. Homogeneity is characterized by range; using SD might be better, less sensitive to outliers than range.
d. Line 326. Is multiple trunks on trees also an expression of patchiness? Certainly a vast difference in forests on this point.

Reviewer 2 ·

Basic reporting

I. Basic Reporting

English language and style:
The language was clear and unambiguous throughout the work however, there were a few spots where improvement was needed. There may be others, but these were most obvious. Please correct this and look for additional examples.

• Line 111 reads as if you are saying Acacia koa is invasive. While I understand that you are paraphrasing referenced information, it should be rewritten so the reader knows A. koa is endemic to the Hawaiian archipelago.
• Line 127 is awkwardly phrased and should be rewritten.
• Line 143 is the beginning of a single sentence paragraph. Please break into two sentences.
• Line 322 and Line 375 are fonts other than the one for the rest of the manuscript. Please correct.

Introduction and background:
The introduction was well-written but perhaps a bit too reference laden. The authors cite 74 references with ~30% (22) of them having been published in 2015 or later. Web of Science showed six other references, well-suited to the topic. Two of these, Bashkin and Binkley 1998 and Lawson and Pike 2017 are very closely aligned to the study rather than a single section as with many of the currently cited references. Please double-check that you are using the very best references for the work you are performing.

1. 1998, Bashkin, MA; Binkley, D. Changes in soil carbon following afforestation in Hawai’I, http://dx.doi.org/10.2307/176582
2. 2003, Osher, LJ; Matson, PA; Amundson, R. Effect of land use change on soil carbon in Hawai’I, http://dx.doi.org/10.1023/A:1026048612540
3. 2007, Chadwick, OA; Kelly, EF; Hotchkiss, SC; Vitousek, PM. Precontact vegetation and soil nutrient status in the shadow of Kohala Volcano, Hawai’I, http://dx.doi.org/10.1016/j.geomorph.2006.07.023
4. 2007, Sandquist, DR; Cordell, S. Functional diversity of carbon-gain, water-use, and leaf-allocation traits in trees of a threatened lowland dry forest in Hawai’I, http://dx.doi.org/10.3732/ajb.94.9.1459
5. 2008, Cordell, S; Sandquist, DR. The impact of an invasive African bunchgrass (Pennisetum setaceum) on water availability and productivity of canopy trees within a tropical dry forest in Hawai’I, http://dx.doi.org/10.1111/j.1365-2435.2008.01471.x
6. 2017, Lawson, SS; Pike, CC. Stable isotope ratios and reforestation potential in Acacia koa populations on Hawai'I, http://dx.doi.org/10.15287/afr.2017.805

Structure and conformity:
This manuscript is written in the proper journal format and the sections are clearly delineated and descriptive.

Figures:
Each of the figures is appropriate for the work and add to the understanding of the research. Figures 1c, d; while clearly outlining the general sites should have been magnified so that greater detail within the sites can be noted. Placement of the green triangles obscures the view and makes it appear as if the remnant forest (RK) was much denser than the koa plantation (KP). Also, Figures 1e, f are providing visualization of the differing densities of kikuyu grass though it appears as if the angle of Figure 1f is such that the grass understory is barely seen in the photo. Do you have another angle or shot? Please edit.
Figure 2 is clear and does a good job of summarizing the stable isotope work. Should there be asterisks over Figure 2a, soil and Figure 2c, Rubus spp.? The straight line representation of the SEM indicates that this is possible, but it is not mentioned in the text.
Figures 3 – 5 are clear.

Tables:
Why are Supplementary Table 2, and Supplementary Tables 3 both referenced before Supplementary Table 1? Please correct.

Raw data:
The link provided in the manuscript to the Github page does, in fact, take you to the site for the raw study data.

Experimental design

II. Experimental Design

Scope:
This manuscript topic is within the Aims & Scope of the journal (https://peerj.com/about/aims-and-scope/).

Research question:
The research question is defined in several multiple of the work (Abstract, Introduction) however, each time it seems to be slightly different. In the Abstract, the research question is “…outplanting of N2-fixing trees produces different biogeochemical landscapes than those observed in reference ecosystems, thereby influencing plant-soil interactions that may influence restoration outcomes…”. The Introduction states “…We hypothesized that the higher density and more even distribution of A. koa in the plantation compared to the remnant forest will result in plantations with higher and spatially more homogeneous N inputs from BNF than remnant forests…”. While similar, these are two different lines of thought. I would clearly demarcate the research question in the Abstract, the Introduction, and the Conclusion and make sure they align. Perhaps use a numeric system and include both questions. For example, the aims of this project are to 1) … and 2)…

Investigation:
The investigation was good and captured a lot of information but there were some missed opportunities. For example, soil and leaf material was captured from these two sites but, between them is an expanse of bare ground (lacking trees) a considerable distance from both sets of trees. Collection of soil samples there would be appropriate for a pseudo base level rather than a simple comparison of plantation and remnant populations. Mention was made of the presence of C4 grasses within the plantation, but it was not sampled (Line 194) as the remnant population lacked substantial grasses. Examination of the grass may have helped tease apart variation between the two sites. Also, as leaf, soil, and twig stable isotope values are all different, collecting twigs while at the site would provide a complete picture.
Overall, the study results were as expected based on previous work and does parallel the studies referenced.

Methods:
I had several concerns about the Materials and Methods section. The authors do a great job explaining where the sites are in reference to each other and their location on the Big Island. The main point of the study was to look at a remnant forest parcel and compare to a plantation parcel. It is noted the plantation trees are 30 years old, but no estimates are made for the remnant forest. Koa reach maturity ~30 but, it is likely those in the remnant forest are much older and much less efficient at carbon assimilation, nitrogen fixation, etc. Did you take that into account? How were the plantation trees planted? Were they dropped into auger holes with a little fertilizer on the bottom? Were they pulled from greenhouse pots and planted with “greenhouse” soil? Was there scarification of the ground (which would uncover hundreds of thousands of grass seeds and could contribute to the wayward growth) before planting? Was there anything in the area before the plantation was planted? Feral cattle and pigs? A clear but concise description of the site would be beneficial. Were these plantation trees grown from seed found in this growth zone? Related genetically?
• Line 182 states samples were “…collected at a spatial resolution relevant for detection of BNF…” What resolution was that exactly?
• Line 200 states three leaves were “…collected and pooled from each tree in the canopy…”. Where in the canopy were these leaves collected (upper canopy, lower canopy, interior, exterior)? Was this standardized for each tree?
A considerable amount of text is used to explain the R packages used but it is not clear if any of the R packages were customized. For example, adjustment to the number of neighbor samples used for interpolation would dramatically alter the map images, as would variation in output classifications. Were there adjustments and, if so, did they alter the output.

Validity of the findings

III. Validity of Findings

Robustness:
The biological replicates described here are acceptable and the rationale for leaf sample pooling was clear. If it is assumed that the plantation trees are younger and still on the upward growth trend, and the remnants are declining, how do you think your interpretation of the results differ? With this in mind, I found many of the results to be as expected based solely on site conditions and tree age.

Additional comments

IV. General Comments

Direct comparison of 2 sites is acceptable but a third site devoid of all koa trees between the two was a missed opportunity.
It is also noted the authors were careful to note immature A. koa trees were in the “thickets adjacent to the planted trees” but how close were they? Figure 1 could have included an addition icon to depict immature tree locations and proximity to site borders. It is mentioned these trees may be contributing to higher N2 fixation rates but nothing more is said about them. Were there immature A. koa at the remnant site?
My conclusions made regarding Figure 4 are a little different that the text. The text indicates “…numerous low 15N areas…” but the map shows two low spots (1, low; 1, very low) at the plantation site. At the remnant site, There are three low spots (1, low; 2, very low). The two very low spots are on opposite corners of the southeastern quadrant and the single low spot resides in the middle. These areas cover the entire plot bottom and likely balloon outward to cover as much or more area than the plantation site.
Could this be just luck? I would shift my remnant quadrant south and examine samples currently outside of the site to see if that swathe of low 15N continues.

This study is a nice look at differences in plantation-grown and remnant forest trees. It does not however, cover the entire island or even multiple areas where reforestation efforts have taken place. There are numerous locations on the island where natural regeneration is lush with those grasses only seen here in large amounts at the plantation site. I think there were many missed opportunities to maximize the information collected.
1. Lack of twig collection.
2. Lack of multi-region comparison (Kona side vs Hilo side)
3. Lack of in-depth knowledge of the plantation tree origins came from or how they were planted.
4. Lack of a clear plan to move the project forward. The conclusions indicate “…future efforts might include a greater diversity of fixing and non-N2-fixing canopy tree species to obtain greater variation in the understory and soil conditions typical of remnant or intact forests.” What about capturing measurements closer to Mauna Loa or Mauna Kea where vog (volcanic organic gases) influences growth? This would be a startling contrast to observe as reforestation efforts have been attempted there as well.

Reviewer 3 ·

Basic reporting

The manuscript written by Wall et al. studied the isoscapes of remnant and restored Hawaiian montane forests. The authors extensively measured the carbon and nitrogen isotope ratios of soils and leaves contrasting two locations. I appreciate the authors’ effort to make the isoscapes of mesh sampling ranging 20 m × 35m.
However, the authors need to validate the advantage of applying this method, although the authors mentioned the validity in lines 296-302. To my knowledge, it is common to compare “multiple plots” in comparing two types of forests. The authors applied “within plots” comparisons between two types of forests. As the authors also admitted in the text, the spatial scale was 3-5 meters, and it is advantageous to find out the patterns in the scale. If the authors wanted to find out a larger scale of ecosystem structure, like BNF or carbon input of plants to the soil, it seems it is more advantageous to compare multiple plots between KP and RK treatments.

Experimental design

Lines 236-239: The authors cited a protocol (Gebauer & Schulze, 1991). However, more details are needed. Especially, the authors need to indicate the working standards (two-point calibration) that were used to correct measured values to an international scale (Paul et al. (2007) Rapid Commun. Mass Spectrom. 21: 3006–3014).

Validity of the findings

The spatial analysis of the isoscapes was appropriate, and the statistical method applied to the manuscript should benefit the audience.

Additional comments

It is confusing to show two types of expressions in supplementary figs and tables. For example, Supplementary Figure 1 should be Figure S1, which is consistent with Table S1. Change all related expressions.
Line 128 “tree species” repeated twice.
Line 311 The authors should cite Table S1.
Line 451 “δ13C value” increases, which is equivalent to “13C is enriched”.

---

## Round 0.2 · accepted · Accept

Thanks for thoroughly addressing the reviewer feedback.

Reviewer 2 ·

Basic reporting

I. Basic Reporting

English language and style:
The language was clear and unambiguous throughout the work however, there were a few spots where improvement was needed. There may be others, but these were most obvious. Please correct this and look for additional examples.

• Line 127 is awkwardly phrased and should be rewritten.

My version of the manuscript reads:


…forests provide are… is the part that is awkward and remains in the text.

All others comments were addressed and I authors’ responses were very much appreciated. Once the above minor change is made (no need to send the paper back to me), I enthusiastically support publication of this work.

Experimental design

No issues

Validity of the findings

No issues

Additional comments

No issues